# Aquaporin-4 in Neuromyelitis Optica Spectrum Disorders: A Target of Autoimmunity in the Central Nervous System

**DOI:** 10.3390/biom12040591

**Published:** 2022-04-17

**Authors:** Yoichiro Abe, Masato Yasui

**Affiliations:** 1Department of Pharmacology, Keio University School of Medicine, Tokyo 160-8582, Japan; 2Keio University Global Research Institute, Tokyo 108-8345, Japan

**Keywords:** aquaporin-4 (AQP4), neuromyelitis optica spectrum disorders (NMOSD), NMO-IgG, Orthogonal arrays of particles (OAPs), astrocytes

## Abstract

Since the discovery of a specific autoantibody in patients with neuromyelitis optica spectrum disorder (NMOSD) in 2004, the water channel aquaporin-4 (AQP4) has attracted attention as a target of autoimmune diseases of the central nervous system. In NMOSD, the autoantibody (NMO-IgG) binds to the extracellular loops of AQP4 as expressed in perivascular astrocytic end-feet and disrupts astrocytes in a complement-dependent manner. NMO-IgG is an excellent marker for distinguishing the disease from other inflammatory demyelinating diseases, such as multiple sclerosis. The unique higher-order structure of AQP4—called orthogonal arrays of particles (OAPs)—as well as its subcellular localization may play a crucial role in the pathogenesis of the disease. Recent studies have also demonstrated complement-independent cytotoxic effects of NMO-IgG. Antibody-induced endocytosis of AQP4 has been suggested to be involved in this mechanism. This review focuses on the binding properties of antibodies that recognize the extracellular region of AQP4 and the characteristics of AQP4 that are implicated in the pathogenesis of NMOSD.

## 1. Introduction

Neuromyelitis optica spectrum disorder (NMOSD) is an autoimmune inflammatory disease of the central nervous system (CNS) that mainly affects the optic nerves and spinal cord [1]. It has long been regarded as a variant of multiple sclerosis (MS). However, in 2004, a specific autoantibody named NMO-IgG, which recognizes the extracellular region of a six-transmembrane water channel aquaporin-4 (AQP4), was identified [2,3]. Since then, it has been suggested that the disruption of astrocytes by complement-dependent cytotoxicity (CDC) induced by NMO-IgG binding to AQP4 is the initial event of NMOSD, identified by using in vitro, ex vivo, and in vivo models [4,5,6,7,8,9,10,11] as well as by analyzing patients’ tissues [12,13]. Recently, mechanism-based therapeutics for NMOSD, including an interleukin 6 inhibitor (Satralizumab), a complement blocking therapy (Eculizumab), and B-cell depletion (Inebilizumab), have been approved worldwide. Progress in the development of new therapeutics for NMOSD based on the mechanism clarified in the past two decades is described in detail in the latest reviews [14,15,16]. However, the detailed mechanisms that link the binding of NMO-IgG to AQP4 on astrocytes, the activation of the classical complement pathway, and/or the complement-dependent disruption of astrocytes with demyelination and neuronal damage remain unclear. Verkman and colleagues demonstrated complement-dependent bystander injury in neurons and oligodendrocytes using in vitro co-culture systems [17,18]. Additionally, antibody-induced cellular cytotoxicity (ADCC) has been suggested as a mechanism that injures neurons, oligodendrocytes, and astrocytes [6,10,11,19,20,21,22,23,24]. In addition to its destructive effect on astrocytes, a distinct mechanism involving excitotoxicity to neurons and oligodendrocytes has been proposed. In this mechanism, the binding of NMO-IgG to AQP4 induces its endocytosis, which causes concomitant loss of excitatory amino acid transporter 2 (EAAT2), followed by disruption of glutamate homeostasis [25,26,27]. The binding of NMO-IgG to AQP4 also reduces glutamine synthetase activity, an astrocytic enzyme that converts glutamate to glutamine, which participates in the local accumulation of glutamate [26]. The binding of NMO-IgG also induces the production of a variety of immune factors in astrocytes, including cytokines, chemokines, and complement components [28], which may recruit and activate immune cells, including neutrophils, eosinophils, and microglia, which are harmful to myelin and neurons [19,21,29]. In any case, the binding of NMO-IgG to astrocytic AQP4 is a key event in the pathogenesis of NMOSD. In this review, we focus on the binding properties of antibodies that recognize the extracellular region of AQP4 and the characteristics of AQP4 that are implicated in the pathogenesis of NMOSD.

## 2. Discovery and Characteristics of AQP4

AQP4 was first cloned in 1994 from rat lung [30] and brain [31] cDNA libraries. Later, AQP4 was found to be expressed in particular cells in a variety of organs, including principal cells of the kidney collecting duct; parietal cells of the stomach; crypt cells in the small intestine; fast-twitch fibers of skeletal muscle; Müller cells of the retina; Hensen’s cells and inner sulcus cells in the inner ear; bronchial, tracheal, and nasopharyngeal epithelium; and satellite glial cells of sensory ganglia in the peripheral nervous system [32,33,34,35,36,37,38,39,40,41,42,43]. In the CNS, it is densely expressed in subpial and perivascular astrocytic end-feet [38,44] and moderately expressed in a subpopulation of ependymal cells [33,44]. Among the 13 mammalian AQPs (AQP0–AQP12), it is the predominant water channel in the central nervous system and plays an important role in fluid transport in the brain [45].

Similar to other AQPs, the minimum unit of the arrangement of AQP4 in the membrane is a tetramer [46,47] (Figure 1D). Importantly, it has been demonstrated that AQP4 tetramers further form supramolecular aggregates known as orthogonal arrays of particles (OAPs) [48,49,50] (Figure 1D), which have been observed using freeze-fracture electron microscopy since the 1970s. Formation of the higher-order structure of AQP4 has been confirmed biochemically using two-dimensional blue native (BN)/sodium dodecyl sulfate polyacrylamide gel electrophoresis (SDS-PAGE) [51,52].

AQP4 has two dominant isoforms: M1 and M23 [31,53,54,55] (Figure 1C). The difference between the two isoforms is the existence of an extra 22 amino acids at the intracellular N-terminus of M1 isoforms; Met^23^ of the M1 isoform corresponds to the initiation codon for M23 (Figure 1C). When M1 and M23 are simultaneously expressed, AQP4 forms relatively smaller OAPs than those formed by M23 alone [56,57] (Figure 1D). M1 and M23 randomly associate with each other to form heterotetramers [46,58]. These results suggest that the N-terminal 22 amino acids of M1 prevent the formation of OAPs, and that the expression ratio between M1 and M23 regulates array size [56,57]. Suzuki et al. demonstrated that Cys^13^ and Cys^17^ of AQP4 M1 are palmitoylated (Figure 1C), and that the substitution of these cysteines with alanine results in the formation of OAPs [58,59,60]. In contrast, several amino acids at the N-terminus of the M23 isoform are involved in the formation of OAPs via hydrophobic peptide–peptide interactions [61]. Rossi et al. demonstrated that OAPs are not formed until AQP4 exits the Golgi apparatus and reaches the plasma membrane, probably because of the presence of a tight membrane curvature in the transport vesicles [62]. Recent analyses of M23-AQP4 null mice have revealed that to fully control the levels of M1 as well as total AQP4 protein, the existence of the M23 isoform or formation of OAPs is required, and the ratio between both isoforms is regulated by a mechanism that probably depends on cell type [63,64]. 

In most cases, AQP4 shows polarized plasma membrane distribution, such as at the perivascular end-feet of astrocytes and basolateral membrane of epithelial cells [32,33,34,36,38,40,41], where the dystrophin complex is localized [65,66,67]. The coexistence of AQP4 with the dystrophin complex has also been demonstrated using BN/SDS-PAGE [68,69], confirming that AQP4 is incorporated into the dystrophin complex. Anchoring AQP4 in the dystrophin complex in astrocytes and skeletal muscle depends on α-syntrophin and is probably achieved by the interaction between the PSD-95-Discs large-ZO1 (PDZ) domain of α-syntrophin and the three C-terminal amino acids, a putative PDZ-binding motif of AQP4 [70,71].

## 3. Production of AQP4 Isoforms at a Transcriptional Level

The open reading frame (ORF) of mammalian AQP4 is divided into five coding exons (exons 0–4, Figure 1A) [54,72,73]. As shown in Figure 1A, AQP4 has multiple transcriptional start sites. The initiation codon for the M1 isoform is located in exon 0; therefore, only a transcript containing exon 0 produces M1 (Figure 1A,B). In rats, there is another minor isoform of AQP4 called AQP4 Mz or AQP4e, which is encoded by a transcript starting from an alternative transcriptional start site upstream of exon 0 (Figure 1A) [72]. This transcript adds 41 amino acids to the N terminus of M1. AQP4 Mz has characteristics similar to those of M1 in terms of water permeability and the inability to form OAPs [72,74]. Regarding M23, initial publications have demonstrated that it is encoded by a transcript starting from another transcriptional start site located immediately upstream of exon 1 (Figure 1A) [30,31,53,54,73]. Subsequently, two other transcripts encoding M23 were identified. One is AQP4 M23X starting from non-coding exon X located between exons 0 and 1 [72,75] (Figure 1A). The other is M23A, which has a distinct five prime untranslated region (5’-UTR) consisting of four non-coding exons (exons A, B, C, and D) located upstream of exon 0 with a variety of combinations by alternative splicing [76] (Figure 1A,B). These non-coding exons are spliced into exon 1, in which the initiation codon for M23 is located (Figure 1B). 

It is conceivable that tissue-specific regulatory mechanisms for the usage of multiple transcriptional start sites described above contribute to the regulation of the ratio of expression between M1 and M23; however, this is yet to be identified. We identified a cluster of *cis*-elements (~440 bp) responsible for the transcription of AQP4 in astrocytes, approximately 2 kb upstream of exon 0, indicated as the astrocyte-specific enhancer (ASE) in Figure 1A, which contains an eight-base consensus binding motif for Pit-1/Oct/Unc-86 (POU) transcription factors (ATGCTAAT) [77,78]. In contrast, Sepramaniam et al. demonstrated that microRNA (miR)-130a binds to the human AQP4 gene approximately 600 bp upstream of the transcriptional start site of M1 mRNA and suppresses activity of the M1 promoter [79]. 

In addition, some nonfunctional isoforms produced by alternative splicing—which skips either exon 2 or 3—were found in rats and humans [72,80] (Figure 1B).

## 4. Production of AQP4 Isoforms at a Translational Level

The M23 isoform can also be produced from M1 mRNA by a leaky scanning mechanism [81] probably because the sequence surrounding the initiation codon for M23 is a better match for the Kozak consensus sequence for translation in eukaryotes (R^−3^NN*AUG*G^+4^) than that of M1 (Figure 1B,C). Additionally, the 5’-UTR of M1 mRNA plays a role in enhancing the translation of M23 from M1 mRNA in rats and humans [82]. In this mechanism, translation of a short out-of-frame upstream ORF (uORF) is initiated with a suboptimal upstream AUG located in the 5’-UTR of M1 mRNA and is terminated at a stop codon, which overlaps with the initiation codon for M1 [82]. The dimensions of the uORF, as well as the distance between the stop codon for the uORF and the initiation codon for M23, are consistent with the ribosome re-initiation mechanism [82] (Figure 1B,C). Although the 5’-UTR of mouse M1 mRNA does not contain the uORF [82], the addition of the 5’-UTR of mouse M1 mRNA to the M1 ORF increases the translation of M23 from the expression construct in Chinese hamster ovary (CHO) cells [83]. One possible explanation for this phenomenon is the formation of a potential internal ribosome entry site, as demonstrated by in silico prediction of the secondary structure of mouse M1 mRNA [64]. 

Translational read-through results in additional isoforms with a 29-amino-acid-extended C-terminus (Figure 1C, green line), called AQP4 M1ex and M23ex [84]. Although AQP4ex accounts for approximately 10% of total AQP4, it plays an important role in its polarized localization at astrocytic end-feet via a strong interaction with α-syntrophin [85,86]. 

It has also been suggested that miR-320a participates in the regulation of translation of AQP4 by changing the level of AQP4 through interaction with the 3’-UTR of its mRNA, both in physiological and pathophysiological conditions [87]. More recently, DEAD-box RNA helicase 17 was found to play a role in regulating total AQP4 expression by inhibiting translation in astrocytes [64].

## 5. Posttranslational Modification of AQP4

We have demonstrated that endogenous AQP4 is constitutively phosphorylated in primary cultured mouse astrocytes by metabolic labeling of cells with H_3_^32^PO_4_, followed by immunoprecipitation with an anti-AQP4 antibody [88]. Exogenously transfected myc-tagged human AQP4, in which four serine and threonine residues (Ser^276^, Ser^285^, Thr^289^, and Ser^316^; putative phosphorylation sites for protein kinase CK2) were changed to Ala, was not labeled with ^32^P in astrocytes [87], indicating that the constitutive phosphorylation of AQP4 occurs in least at some of these Ser/Thr residues located in the C-terminal domain. Assentoft et al. confirmed the constitutive phosphorylation of Ser^276^, Ser^285^, and Ser^316^ in rat AQP4 by mass spectrometry [88]. In addition to these serine residues, phosphorylation has been detected at Ser^315^ (Gln^315^ in human AQP4), Ser^321^, and Ser^322^ [89].

## 6. Characteristics of NMO-IgG

Multiple studies have demonstrated that NMO-IgG does not directly interfere with the water permeability of AQP4 [90,91,92], probably because NMO-IgG cannot access all four monomers—each of which has a water pore—because of the large size of immunoglobulin compared to the tetrameric composition of AQP4, as Rossi et al. pointed out [91]. It should be noted that endocytosis of AQP4 induced by binding of NMO-IgG reduces water movement across the plasma membrane [93]. 

Although the epitope of NMO-IgG is restricted to the extracellular region of AQP4, which consists of three loops connecting six transmembrane domains, there are some variations in its binding properties against AQP4, indicating that NMO-IgG is a group of antibodies recognizing distinct parts of the extracellular region of AQP4 and exists as a mixture thereof in a patient’s plasma [6,90,92,94,95,96,97]. In most cases, NMO-IgG is unable to recognize denatured AQP4, suggesting that a majority of NMO-IgG recognizes the three-dimensional structure of AQP4, which makes it difficult to identify its accurate epitope [6,90,92]. This is supported by the results obtained by Pisani et al., in which introducing mutations into more than one loop affected the binding capacity for NMO-IgG in sera from patients [96]. It has been demonstrated that most NMO-IgGs preferentially bind to AQP4 incorporated into OAPs, although the primary sequence of the extracellular region between M1 and M23 is the same [90,92,95,96,98]. The formation of AQP4 OAPs further enhances multivalent binding of complement C1q to clustered NMO-IgG [99,100]. To the best of our knowledge, most IgG fractions derived from NMO-IgG-seropositive Japanese patient sera less frequently recognize rodent AQP4 [92,94], which may be one of the problems in establishing a good rodent model for NMOSD using NMO-IgG [5,7,23].

## 7. Development of Monoclonal Anti-AQP4 Antibodies

Monoclonal antibodies against the extracellular domains of AQP4 were first established by cloning cDNAs encoding heavy and light chains from CD138+ plasmablast clones isolated from the patient’s cerebrospinal fluid [6]. Verkman and colleagues introduced amino acid substitutions into the fragment crystallizable (Fc) region of a recombinant antibody, rAb53, to eliminate the functionality of CDC and ADCC [101]. They demonstrated that the “non-pathogenic” antibody (aquaporumab) prevents CDC and ADCC in vitro and reduces NMO-like lesions in an in vivo mouse model [9] as well as an ex vivo spinal cord slice model [10], suggesting that preventing the binding of NMO-IgG to AQP4 is a potential therapeutic option [101]. Recently, mutations were introduced into the fragment antigen binding (Fab) region of aquaporumab to achieve ~8-fold greater affinity than the original antibody [102].

We independently established monoclonal antibodies that recognize the extracellular domains of AQP4. For this purpose, we utilized the baculovirus display method [103]. Using this method, we first constructed a budded baculovirus expressing human AQP4 M23 on the envelope and immunized BALB/c mice expressing gp64, a highly immunogenic baculovirus protein expressed on the envelope, with the AQP4-expressing virus [92,104]. Three antibodies were used, as shown in Table 1. While two antibodies (C9401 and D12092) exclusively recognize human AQP4 isoforms, the other (D15107) can also bind to the M23 isoform of mouse AQP4 with weaker affinity [92,104]. Although all three antibodies recognize both M1 and M23 isoforms, they preferentially bind to the M23 isoform as most NMO-IgGs do [104], indicating that OAP formation of AQP4 contributes to the binding of these antibodies. We then created chimeric D15107, in which the constant regions of both heavy and light chains were changed to those of human IgG1 and κ, respectively [104]. The constant region of human IgG1 contains several mutations that eliminate CDC and ADCC functionality. Chimeric D15107 prevents NMO-IgG-induced CDC in CHO cells expressing human AQP4 in a dose-dependent manner [104].

We also established rodent-selective antibodies against the extracellular domains of AQP4, E5415A, and E5415B (Table 1) by immunizing gp64-transgenic BALB/c mice with an AQP4 knockout background [43,105] with budded baculovirus expressing mouse AQP4 M23 [83,106]. Interestingly, E5415A binds to both M1 and M23 of AQP4, whereas E5415B only recognizes the OAP-forming M23 isoform [83].

## 8. Binding Properties of Antibodies against the Extracellular Domains and Endocytosis of AQP4

In general, as mentioned above, NMO-IgG preferentially binds to AQP4 incorporated into OAPs. Pisani et al. speculated that when AQP4 tetramers are incorporated into OAPs, the interaction between the extracellular loops of two adjacent tetramers changes the 3D configuration of the three loops in each monomer to one favorable for NMO-IgG binding [96]. Because they used patient sera, they did not perform quantitative analyses. Crane et al. addressed this issue using monoclonal antibodies cloned from patient plasmablasts against AQP4 expressed in U87-MG cells, a cell line originating from human astrocytoma [95]. One of the antibodies, rAb58, binds to both M1 and M23 with a similar affinity, whereas Ab53 preferentially binds to M23 [95]. Interestingly, the Fab fragment of rAb53 can also distinguish between M1 and M23 [95]. Thus, they concluded that both antibodies bind to AQP4 in a monovalent manner and that while an epitope for rAbe53 is created upon OAP assembly—such as the interface between two tetramers—the binding site for rAb58 is located within a tetramer [94].

We examined the binding properties of antibodies against the extracellular region of AQP4 using rodent-selective monoclonal antibodies, E5415A and E5415B, against mouse AQP4 expressed in fixed CHO cells with various expression levels and ratios between M1 and M23 [83]. We obtained four observations that indicated the binding properties of NMO-IgGs against AQP4. First, E5415B can only bind to cells expressing OAP-forming M23 (Figure 2, middle and bottom), whereas E5415A can also bind to cells expressing M1 alone (Figure 2, top), although the affinity of E5415A for M1 is approximately 10 times lower than that for M23, indicating that OAP formation of AQP4 is advantageous for both antibodies. Second, although both E5415A and E5415B bind to large OAPs expressed in cells expressing M23 alone (Figure 2, bottom) with similar binding capacities, E5415A showed greater binding capacity than E5415B in cells expressing both M1 and M23 (therefore, having relatively small OAPs) (Figure 1D and Figure 2, middle), indicating that E5415A has more binding sites where E5415B is unable to bind. Third, the dissociation of both E5415A and E5415B from OAPs after washout and further incubation in buffer without the antibody occur very slowly; on the other hand, the bound antibodies dissociate very rapidly from OAPs when an excess of the antibody exists in the buffer, indicating that there are two modes of dissociation for E5415A and E5415B. Fourth, dissociation of E5415A from cells expressing M1 is also accelerated by the addition of an excess of the antibody in the buffer, and that even an excess of E5415B can enhance dissociation of E5415A from AQP4 tetramers. This indicates that dissociation of E5415A from AQP4 without OAPs also has two modes, and that E5415B, which is unable to bind in the absence of OAPs, can interact with an AQP4 tetramer. Based on these observations, we propose a model in which both antibodies bind to AQP4 in a bivalent manner (Figure 3). In this model, the monovalent binding of both E5415A and E5415B is very weak; however, once bivalent binding is established, the antibodies will not dissociate easily. The transition from bivalent to monovalent occurs very rapidly; however, OAPs enhance the rapid transition back from monovalent to bivalent, probably because the distance between two tetramers in OAPs is suitable for bivalent binding of the antibodies.

Several groups have observed that some antibodies against AQP4 induce endocytosis of AQP4 in cell lines exogenously expressing AQP4 and in primary cultured astrocytes [4,19,25,83,104]. Hinson et al. has demonstrated that M1 is rapidly internalized upon binding of NMO-IgG, whereas M23 is not, using HEK293 cells stably expressing either M1 or M23 [100]. Fluorescence-labeled E5415A was administered to CHO cells expressing either M1 or M23 alone, and it was observed that the fluorescence-labeled E5415A rapidly internalized and accumulated in the intracellular compartment, indicating that both M1 and M23 undergo endocytosis [83]. In cells expressing M1 alone, the level of AQP4 protein was drastically reduced after 24 h of incubation with E5415A and was restored to the basal level if bafilomycin A1, an inhibitor of the vacuolar-type proton pump—which increases the pH value in lysosomes and inhibits protease activity—was simultaneously added, indicating that antibody-bound AQP4 was transported to lysosomes and degraded [83]. On the other hand, the level of M23 did not change after 24 h incubation with E5415A or E5415B, although we observed rapid internalization of fluorescence-labeled antibodies observed by live imaging, as mentioned above [83]. However, the addition of bafilomycin A1 increased the level of AQP4 over basal levels regardless of the addition of antibodies, suggesting that turnover of M23 occurs more rapidly than M1, and that the binding of antibodies does not accelerate it [83]. In cells expressing both M1 and M23, E5415A also enhanced endocytosis of AQP4, whereas E5415B did not—although E5415B can also bind to these cells [83]. The difference between the two antibodies is that while E5415A forms large clusters of AQP4 by cross-linking more than two tetramers and OAPs, E5415B does not [83] (Figure 2, top and middle, respectively). To form a cluster of tetramers, at least two E5415A molecules must bind to a tetramer (Figure 2, top, white circle). These observations suggest that the cluster formation of AQP4 via antibody binding is a trigger to accelerate endocytosis of AQP4, and the formation of extremely large arrays may have the same effect. Interestingly, the binding capacity of D12092 to human AQP4 M1 expressed in live CHO cells for one hour at 37 °C increased compared with that of fixed cells [104], probably because of the rapid diffusion of M1 tetramers in live cells [107], thus increasing opportunities to locate tetramers favorable to the bivalent binding of D12092. In contrast, the binding capacity of D15107 to live AQP4-expressing cells decreased at high concentrations due to accelerated endocytosis of AQP4 [104]. These observations suggest that D15107 can cross-link more than two tetramers, whereas D12092 cannot.

## 9. Effects of Rodent-Selective Monoclonal Antibodies in In Vivo Models

We have demonstrated that a single intraperitoneal injection of E5415A can evoke (1) severe clinical exacerbation and produce lesions resembling NMO, including extensive loss of astrocytic markers such as AQP4, EAAT2, and glial fibrillary acidic protein (GFAP), as well as (2) perivascular deposition of IgG and C5b-C9 in the spinal cord, brainstem, and optic chiasm in rats with myelin basic protein (MBP)-induced experimental autoimmune encephalomyelitis (EAE) in which activated T cells increase the blood–brain barrier permeability [23]. Rats receiving a high dose of E5415A (0.1–1 mg) showed neutrophil infiltration at the lesion edge, which is associated with demyelination and axonal damage, suggesting a mechanism involving both CDC and ADCC [23].

Hillebrand et al. demonstrated that daily injection of E5415A (1 mg) for up to 5 days induces NMO-like pathology, including loss of AQP4, complement activation, and immune-cell infiltration in rats, even in the absence of T-cell responses [11]. They showed three distinct sites for E5415A to enter the CNS: the area postrema, meningeal vessels, and parenchymal vessels. Interestingly, lesions in the area postrema are relatively mild with little complement deposition and few immune cell infiltrations in contrast to those in the subpial/periventricular and perivascular regions [11]. In addition, they also demonstrated that injected antibodies would be absorbed by peripheral organs expressing AQP4—such as the kidneys and stomach—to reduce the concentration of circulating antibodies, which is associated with a decrease in the expression of AQP4 in those organs [11]. 

Chen et al. continuously infused E5415B (1 μg/day) and the IgG fraction from NMOSD patients’ serum (10 μg/day) in the absence of an exogenous complement using an osmotic minipump into the spinal subarachnoid space for 5 to 7 days and demonstrated motor deficits and hind limb paralysis [29]. Although AQP4 and NeuN immunoreactivity were reduced, the terminal membrane attack complex of the complement system was not observed, indicating that neuronal damage was induced without CDC [29]. 

## 10. Concluding Remarks

Considering the unique features of AQP4—the formation of OAPs and the dense localization to astrocytic end-foot surrounding blood vessels—it is reasonable that AQP4 preferentially becomes a target of an autoantibody, even if the affinity for direct interaction between epitope and paratope is weak. Recently, mechanism-based therapeutics for NMOSD, including interleukin 6 inhibitors, a complement blocking therapy, and B-cell depletion have been approved worldwide. Preventing the binding of NMO-IgG to AQP4 is more specific to the disease than the mechanism-based therapeutics listed above—which suppress a wide range of immune systems—and is a potential therapeutic option to treat NMOSD. On the other hand, according to recent studies using in vitro and in vivo models, the binding of antibodies to AQP4 seems to induce cellular responses other than CDC or ADCC, endocytosis of AQP4 followed by disruption of glutamate homeostasis, and stimulation of intracellular signaling pathways to produce a variety of chemical mediators that recruit immune cells, including neutrophils, eosinophils, and microglia, which are harmful to myelin and neurons. If this is the case, antibodies lacking functionality for CDC and ADCC that compete with NMO-IgG for the extracellular region of AQP4 may be harmful. Further studies are necessary to identify the detailed mechanisms underlying the binding of NMO-IgG to AQP4, independent of CDC or ADCC.

## Figures and Tables

**Figure 1 biomolecules-12-00591-f001:**
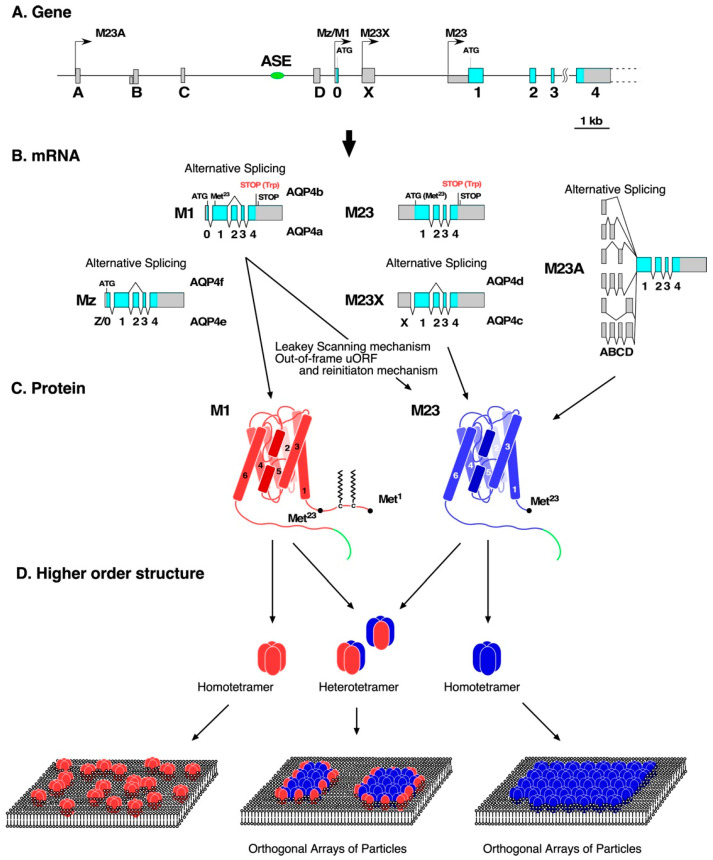
Characteristics of AQP4 expression, translation, and expression on the cell surface. (**A**) A schematic illustration of structure of the AQP4 gene. Transcriptional start sites are indicated with arrows. Exons are represented by boxes. Cording and non-cording regions are indicated in blue and gray, respectively. (**B**) Schematic illustrations of structure of mRNA encoding AQP4. Cording and non-cording regions are indicated in blue and gray, respectively. (**C**) Conformation of M1 (red) and M23 (blue) monomers. A region added to the C terminus of each isoform as a result of transcriptional read-through is represented as a green line. (**D**) Higher-order structures of AQP4; M1 and M23 are indicated in red and blue, respectively.

**Figure 2 biomolecules-12-00591-f002:**
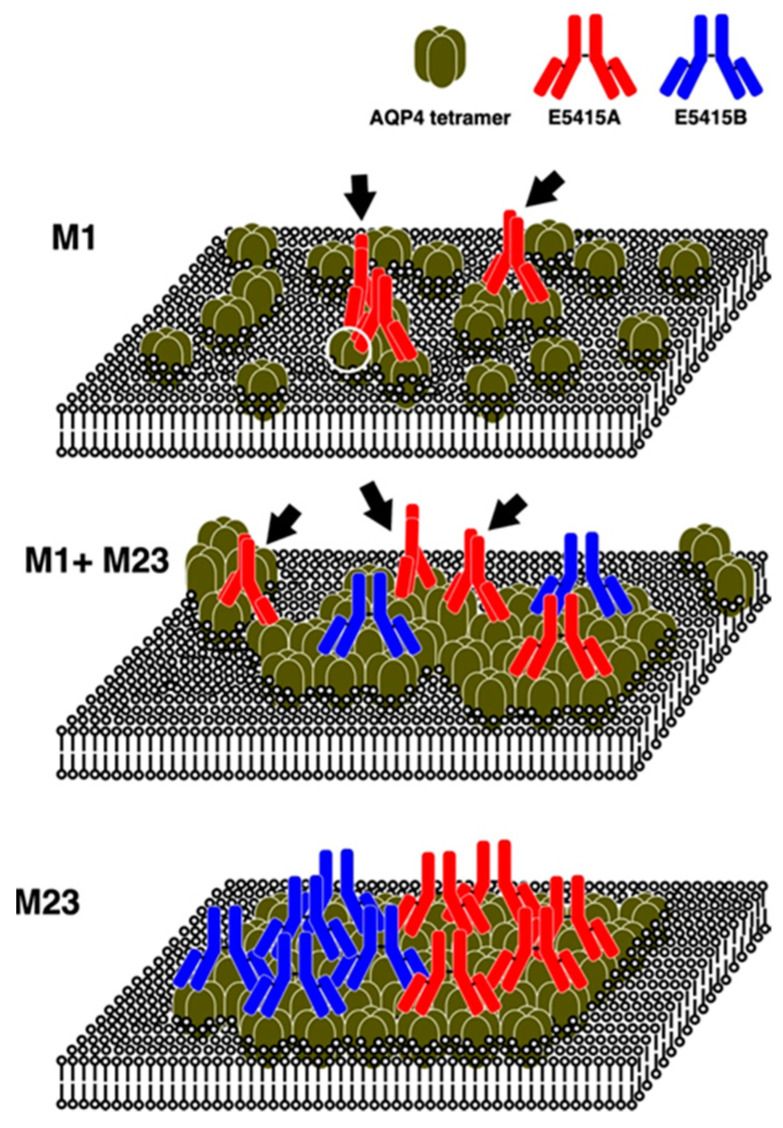
Schematic illustrations of putative binding of E5415A and E5415B against AQP4 on the plasma membrane. E5415A and E5415B are represented in red and blue, respectively. E5415A molecules binding to AQP4 tetramers, which are not incorporated into OAPs, are indicated with arrows. An AQP4 tetramer binding to two E5415A molecules is indicated with a white circle.

**Figure 3 biomolecules-12-00591-f003:**
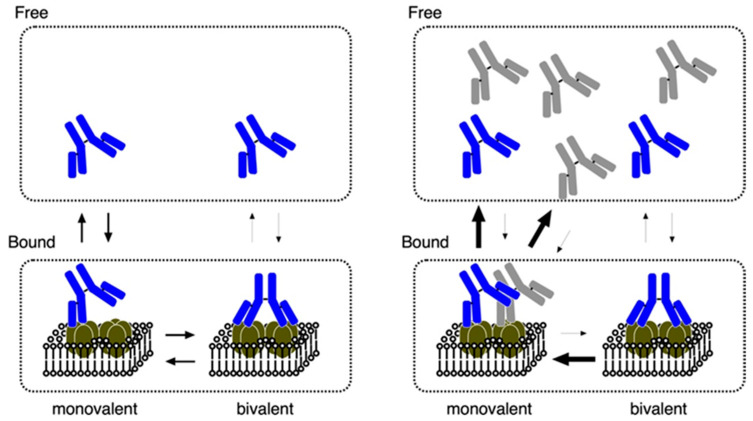
Schematic illustrations of a model for binding modes of E5415A and E5415B to OAPs.

**Table 1 biomolecules-12-00591-t001:** Monoclonal antibodies against the extracellular domains of AQP4.

Clone	Antigen	Subclass	Binding			
			hAQP4 M1	hAQP4 M23	mAQP4 M1	mAQP4 M23
C9401	hAQP4 M23/BV	2b/κ	+	+++	-	-
D12092	hAQP4 M23/BV	2b/κ	+	+++	-	-
D15107	hAQP4 M23/BV	2a/κ	+	+++	-	+
E5415A	mAQP4 M23/BV	2a/κ	ND	+	+	+++
E5415B	mAQP4 M23/BV	2a/κ	ND	-	-	+++

BV, budded baculovirus; ND, not determined; +++, high affinity; +, low affinity; -, no binding.

## Data Availability

Not applicable.

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
