# Peer review of "Aquaporin-4 in Neuromyelitis Optica Spectrum Disorders: A Target of Autoimmunity in the Central Nervous System"

_biomolecules, 2022, doi:10.3390/biom12040591_

Round 1

Reviewer 1 Report

The study examined the general characteristics of AQP4, and offer detail description of the different isoforms of this protein, at transcriptional and translational level, found in tissues and discus the contribution that the binding of antibodies anti-NMO to AQP4 represent in the pathogenesis of NMOSD. It provides comprehensive analysis of how antibodies NMO-IgG, binds to AQP4, and present data regarding in vitro and in vivo models to explain that binding of antibodies to AQP4 seems to produce both, damage by direct implication of complement immune response and damage that seems to be independent of complement. The study is overall very interesting, and the manuscript is well presented and written.

Minor points:

- In line 77, when you say minimum unit of AQP4 is a tetramer…… I think that will be better to say the minimum unit of arrangement in the membrane.

- In line 123, there is a spelling error in the word: non-coding, that appear as non-cording.

- In the paragraph, that start on line 182, where you indicate that multiple studies have demonstrated that NMO-IgG does not interfere with the water permeability of AQP4, I would like to indicate that there are papers that also indicate the contrary. In fact, Netti V. et al (Molecular Neurobiology, 58, 5178-5193, 2021). Demonstrated that in Müller cells, NMO-IgG produce reduction of plasma membrane water permeability because binding of these antibodies to AQP4 produce partial internalization of the protein. Can you discuss on that and consider mentioning this other recent evidence?

Author Response

- In line 77, when you say minimum unit of AQP4 is a tetramer…… I think that will be better to say the minimum unit of arrangement in the membrane.

We changed the sentence in accordance with the reviewer’s suggestion.

- In line 123, there is a spelling error in the word: non-coding, that appear as non-cording.

We corrected the error.

- In the paragraph, that start on line 182, where you indicate that multiple studies have demonstrated that NMO-IgG does not interfere with the water permeability of AQP4, I would like to indicate that there are papers that also indicate the contrary. In fact, Netti V. et al (Molecular Neurobiology, 58, 5178-5193, 2021). Demonstrated that in Müller cells, NMO-IgG produce reduction of plasma membrane water permeability because binding of these antibodies to AQP4 produce partial internalization of the protein. Can you discuss on that and consider mentioning this other recent evidence?

We have excluded indirect functional disturbance of AQP4 by NMO-IgG such as endocytosis of AQP4. However, as the reviewer mentioned, it is reasonable that removal of AQP4 from the cell surface reduces water movement across the plasma membrane. Thus, we changed the sentence “multiple studies have demonstrated that NMO-IgG does not interfere with the water permeability of AQP4...” to “multiple studies have demonstrated that NMO-IgG does not DIRECTLY interfere with the water permeability of AQP4...” and added a sentence, “It should be noted that endocytosis of AQP4 induced by binding of NMO-IgG reduces water movement across the plasma membrane” with citing the paper (Netti V. et al. Molecular Neurobiology, 58, 5178-5193, 2021) as the reviewer recommended.

Reviewer 2 Report

1- What is the role of NMO-IG in other disorders?

2- Why are some NMOSD patients seronegative?

3- Introduction is too long and it is better to divide to some separate sections.

Author Response

1- What is the role of NMO-IG in other disorders?

No diseases in which NMO-IgG plays a pathological role other than NMOSD have been identified. NMO-IgG (IgG against the extracellular domains of AQP4) is specific to NMOSD and therefore, it is used as a reliable maker for definitive diagnosis of this disease.

2- Why are some NMOSD patients seronegative?

A part of NMO-IgG-seronegative patients have anti-MOG IgG. Although the symptoms of patients with anti-MOG IgG are similar to NMO-IgG seropositive NMOSD, pathogenesis of this disease is different and diseases with anti-MOG antibody are recategorized as MOG-IgG associated disorders( MOGAD) very recently.

Pathogenesis of NMOSD patients lacking anti-AQP4 or anti-MOG IgG is yet to be elucidated. However, once the pathogenesis of  the symptoms is identified, it will probably be excluded from NMOSD.

NMOSD will probably become a disease characterized by the existence of anti-AQP4 in the future.

3- Introduction is too long and it is better to divide to some separate sections.

We believe that the length of the introduction is appropriate and are unable to divide the paragraph into multiple senctins.